# Heparan sulfate attachment receptor is a major selection factor for attenuated enterovirus 71 mutants during cell culture adaptation

**Kyousuke Kobayashi**[1], **Katsumi Mizuta**[2], **Satoshi Koike**[1]*

**1** Neurovirology Project, Department of Genome Medicine, Tokyo Metropolitan Institute of Medical Science, Tokyo, Japan, **2** Department of Microbiology, Yamagata Prefectural Institute of Public Health, Yamagata, Japan

* koike-st@igakuken.or.jp

**Data Availability Statement:** All relevant data are within the manuscript.

**Funding:** This work was supported in part by Japan Society for the Promotion of Science (JSPS)

## Abstract

Enterovirus 71 (EV71) is a causative agent of hand, foot, and mouth disease (HFMD). However, this infection is sometimes associated with severe neurological complications. Identification of neurovirulence determinants is important to understand the pathogenesis of EV71. One of the problems in evaluating EV71 virulence is that its genome sequence changes rapidly during replication in cultured cells. The factors that induce rapid mutations in the EV71 genome in cultured cells are unclear. Here, we illustrate the population dynamics during adaptation to RD-A cells using EV71 strains isolated from HFMD patients. We identified a reproducible amino acid substitution from glutamic acid (E) to glycine (G) or glutamine (Q) in residue 145 of the VP1 protein (VP1-145) after adaptation to RD-A cells, which was associated with attenuation in human scavenger receptor B2 transgenic (hSCARB2 tg) mice. Because previous reports demonstrated that VP1-145G and Q mutants efficiently infect cultured cells by binding to heparan sulfate (HS), we hypothesized that HS expressed on the cell surface is a major factor for this selection. Supporting this hypothesis, selection of the VP1-145 mutant was prevented by depletion of HS and overexpression of hSCARB2 in RD-A cells. In addition, this mutation promotes the acquisition of secondary amino acid substitutions at various positions of the EV71 capsid to increase its fitness in cultured cells. These results indicate that attachment receptors, especially HS, are important factors for selection of VP1-145 mutants and subsequent capsid mutations. Moreover, we offer an efficient method for isolation and propagation of EV71 virulent strains with minimal selection pressure for attenuation.

## Author summary

Viruses must overcome various setbacks in a variety of tissues and cells during transmission from the initial replication site to the final target site. To achieve this, RNA viruses employ a strategy to adapt to different environments by creating a diverse viral population using low-fidelity RNA-dependent RNA polymerases. On the other hand, when the

KAKENHI (https://www.jsps.go.jp/index.html),
grant number JP18H02667 to SK and 19K07601 to
KK and by Japan Agency for Medical Research and
Development (AMED, https://www.amed.go.jp/),
grant number 19fk0108084h1101 to SK. The
funders had no role in study design, data collection
and analysis, decision to publish, or preparation of
the manuscript.

**Competing interests:** The authors have declared
that no competing interests exist.

viruses are propagated in clonal cell cultures, *in vitro* adaptation occurs. The viruses may acquire new properties or lose some properties they had *in vivo*. *In vitro* adaptation is often associated with attenuation. Therefore, the selection pressures imposed on viruses replicating *in vitro* and *in vivo* are quite different. It is unclear how this environmental difference affects viral populations. Clinical isolates of EV71 replicate in cultured cells poorly. However, after a few passages, the viruses adapt to this condition and replicate efficiently. In this study, we demonstrate that attachment receptor usage is a major selection pressure for *in vitro* adaptation of EV71 by analyzing the population dynamics of cell culture-adapted viruses. This mechanism appears to be a major mode of attenuation.

## Introduction

Enterovirus 71 (EV71), which belongs to the genus *Enterovirus* of the family *Picornaviridae*, has a single-stranded positive-sense RNA genome encoding a single polyprotein that is flanked by two untranslated regions (UTR), a 5'UTR and a 3'UTR. Polyproteins are comprised of three regions, P1, P2, and P3. The EV71 icosahedral capsid is composed of 60 copies of the capsid proteins (VP1, VP2, VP3, and VP4), which are coded in the P1 region of the viral genome [1]. EV71 is a causative agent of hand, foot, and mouth disease (HFMD), together with other members of the human enterovirus A (EV-A) [2]. HFMD is a mild and self-limiting disease that manifests as vesicular lesions of the hands, feet, and mouth. However, unlike other EV-A serotypes, infection with EV71 is sometimes associated with severe neurological complications. Since the late 1990s, large outbreaks have repeatedly occurred in Asian countries [3–7]. It is still unclear what viral factors and host factors are critical for EV71 neurovirulence.

It is important to identify neurovirulence determinants in the viral genome to understand EV71 pathogenicity. It has been reported that an amino acid substitution at residue 145 of the VP1 protein (VP1-145) plays an important role in EV71 virulence. Viruses that have glutamic acid (E) at VP1-145 (VP1-145E viruses) are virulent in several animal infection models [8–15], whereas viruses that have glycine (G) or glutamine (Q) (VP1-145G and VP1-145Q viruses) are avirulent. However, most EV71 strains that are freshly isolated from infected patients have E at this position [16, 17], suggesting that mutation at this site alone is not sufficient to explain the difference in virulence among the circulating EV71 strains, and that additional determinants exist in the EV71 genome. However, other determinants are poorly reported. For example, amino acid mutations at VP1-170 [18], VP1-97 [19], 3D-73, 3D-362 [20], VP1-164, and 2A-68, [21], as well as nucleotide mutations at 5'UTR-158 [21], -272, -488, -700 [22], -485, and 3'UTR-7408 [20], have been reported as virulence determinants of EV71.

One of the main issues in evaluating EV71 virulence determinants is that the genomic sequence of EV71 changes very rapidly during its isolation and propagation in cultured cells, such as RD-A cells and Vero cells [23, 24]. EV71 freshly isolated from human samples grow very poorly in cultured cells. Nucleotide and amino acid substitutions can occur after only a few passages during the isolation of the virus from human specimens or during recovery of the virus from the infectious cDNA clone, which makes it difficult to obtain accurate results even using reverse genetics. The rapid change in the virus sequence suggests that a strong selection pressure is imposed on the viral population in cultured cells. However, it is unclear what factors cause the selection of certain mutations, and detailed information on how the EV71 population changes in culture is lacking. In the case of RNA viruses, including EV71, replication errors in the viral genomic RNA are induced by their low-fidelity RNA-dependent RNA polymerases. This machinery diversifies the RNA virus population, which plays an important role in survival and adaptation to various environments [25]. Selection pressures on the virus

population in host animals are thought to be quite different from selection pressures in cultured cells. To replicate *in vivo*, viruses must overcome various setbacks in a variety of tissues or cells and must escape detection by the host immune system. Through the mutation and selection process, tissue/cell-specific virus populations are generated, as reported for EV71 and poliovirus [19, 26, 27]. Viruses isolated from clinical samples of infected patients should therefore be adapted to various *in vivo* environments. However, the isolated viruses are usually propagated for characterization in clonal cell cultures, in the absence of an acquired immune system. During this process, cell culture adaptation occurs. It is unclear how this environmental change affects viral populations *in vitro*. Analyzing the population dynamics of viruses during *in vitro* adaptation should reveal useful information about the factors that drive adaptive selection of the virus, and how unwanted adaptation in cultured cells could be avoided.

Virus receptors are considered to be one of the selection pressures for virus selection. EV71 infection is initiated by attachment of the virus to the cell surface, followed by its internalization and the release of viral genomic RNA into the cytoplasm of infected cells, a process called uncoating. We previously reported that human scavenger receptor class B, member 2 (hSCARB2) can support these three steps [28]. All EV71 strains can use hSCARB2 as a receptor [29]. hSCARB2 transgenic (tg) mice are susceptible to EV71 infection, and EV71-infected mice show neurological disease [30]. hSCARB2 binds the south rim of the canyon of the EV71 virion [31], and this binding initiates uncoating at a low pH [30]. However, SCARB2 is a lysosomal protein and is not abundantly expressed on the surface of cultured cells. Therefore, this step can be a bottleneck on EV71 replication. Some EV71 strains also use so-called attachment receptors, including P-selectin glycoprotein ligand-1 (PSGL-1) [32], heparan sulfate (HS) [33], annexin II [34], sialic acid [35], nucleolin [36], vimentin [37], and fibronectin [38]. The attachment receptors can bind to the virus at the cell surface and enhance infection, although attachment receptors alone are not sufficient for establishment of infection because they cannot initiate uncoating of the virion. The amino acid residues near the five-fold axis, which includes VP1-145, determine binding specificity to HS and PSGL-1 [13, 24, 33]. The surface of the VP1-145G and VP1-145Q virion around the five-fold axis is rich in positively charged amino acids [39], allowing for electrostatic interaction with HS and highly sulfated PSGL-1. The negative charge of the E residue at VP1-145 neutralizes the positive surface charge, resulting in decreased affinity to HS and PSGL-1 [24, 39]. The binding specificity of EV71 to other attachment receptors has not been elucidated in detail.

We hypothesized that attachment receptors play an important role in the selection of viral populations *in vitro*. Here, we analyzed the population dynamics of EV71 strains that were initially virulent *in vivo* during cell culture adaptation. We found that EV71, which acquired a mutation in VP1-145, was effectively selected in cultured cells. This mutation caused attenuation of virulent strains. We hypothesized that HS expressed on the cell surface is a major factor for this selection in RD-A cells. We confirmed this hypothesis using HS-deficient, hSCARB2-overexpressing cells. In addition, this mutation further promotes the acquisition of secondary mutations in the EV71 capsid to increase the fitness of the virus in cultured cells. We propose that attachment receptor usage is a major factor for adaptation of EV71 *in vitro*.

## Results

### Viruses carrying VP1-145G and Q mutations selectively replicate during passage of EV71 strains in RD-A cells

RD-A cells and Vero cells are commonly used for the isolation of EV71 from clinical samples of HFMD patients. During the isolation process, EV71 replicates very poorly. A cytopathic effect (CPE) does not always appear immediately, but sometimes appears after several blind

passages. In addition, virus stocks prepared from a virus with a low passage history tend to have a low virus titer. However, after several passages, the CPE appears much earlier and the viral titer reaches high levels. These results suggested that the cell culture conditions required for viral proliferation are quite different from those *in vivo* and that virus fitness under cell culture conditions is very low, indicating that adaptation and selection of the adapted virus must occur to overcome this low fitness during this process.

To identify the mutations selected in cultured cells, we analyzed single nucleotide variations (SNVs) occurring in the EV71 genome after passage in RD-A cells. The 2716-Yamagata-03 (2716-Ymg-03) strain, which is classified into subgenogroup B5, was isolated from an HFMD patient using GMK cells, passaged two generations [16], and passaged one generation in RD-A-overexpressing hSCARB2 (RD+hSCARB2) cells. This stock was used as the starting material (passage-0; p-0) for this experiment. The SI/Isehara/Japan/99 (Isehara) strain, which is classified into subgenogroup C2, was isolated from an HFMD patient and passaged several times before we received it. Although the passage history of this virus is not clear, we constructed an infectious cDNA clone for this strain [13], prepared the virus from the cloned cDNA after transfection of the *in vitro*-transcribed RNA into RD+hSCARB2 cells, and then propagated it by two passages in RD+hSCARB2 cells. This was used as p-0 virus. These viruses were passaged in RD-A cells three times at a low multiplicity of infection (MOI < 0.01). Subsequent generations were designated as p-1, p-2, and p-3. To evaluate mutations arising in each virus population, we performed next-generation sequencing (NGS) on viral RNA from the p-0, p-1, and p-3 virus samples, mapped the reads to the consensus sequences of the p-0 virus, and estimated the abundance of each SNV (Fig 1A and 1B).

In p-1 and p-3 2716-Ymg-03 populations, we identified four mutations with more than 5% abundance at nucleotide positions 186, 2875, 2876, and 6517 of the virus genome (Fig 1A). The nonsynonymous mutations at 2875 and 2876 are located in the same codon, corresponding to VP1-145, where the former is an E-to-Q amino acid substitution and the latter is an E-to-G substitution. In the p-0 population, the VP1-145G mutant was detected at an abundance of approximately 5% and the VP1-145E at 95%, but VP1-145Q was not detected (Fig 1C). After one passage, the VP1-145G virus displaced the VP1-145E population (64.5%), and VP1-145Q appeared (15%). The virus population was completely composed of VP1-145Q virus (100%) after two more passages in RD-A cells (p-3). To exclude the possibility of a double mutation in the same codon (2875 and 2876), which would encode arginine (R), we analyzed the region from 2800 to 2910 by local haplotype analysis (Fig 1C) and detected no E-to-R substitution at VP1-145. The other nonsynonymous mutation at 6517 substituted histidine to tyrosine at amino acid residue 193 of 3D polymerase. This mutation also increased during passage (0%, 8.3%, and 20.9% in p-0, p-1, and p-3, respectively). The deletion of C at position 186 in the 5'UTR was detected in p-0 at an abundance of approximately 12% and was increased in p-1 (59.3%), but it was not detected in p-3.

In the passage experiment with the Isehara strain, two nonsynonymous mutations were detected with more than 5% abundance in the p-1 population, at nucleotide positions 2872 and 4051 (Fig 1B). These mutations lead to amino acid substitutions at VP1-145 and 2B-91, respectively. In the p-3 population, we found four nonsynonymous mutations, at nucleotide positions 1373, 2793, 2872, and 3184, all of which are located in the P1 region and lead to amino acid substitutions at VP2-141, VP1-119, VP1-145, and VP1-249, respectively. The E-to-Q amino acid substitution at VP1-145 was the only mutation detected in both the 2716-Ymg-03 and Isehara populations at a high abundance. In the p-0 population of the Isehara strain, no VP1-145 mutations were detected (Fig 1D). However, after just one passage in RD-A cells, VP1-145Q virus became dominant (92.8% and 99.6% in the p-1 and p-3 populations, respectively). Other mutations also increased during passage but did not reach 100% abundance in

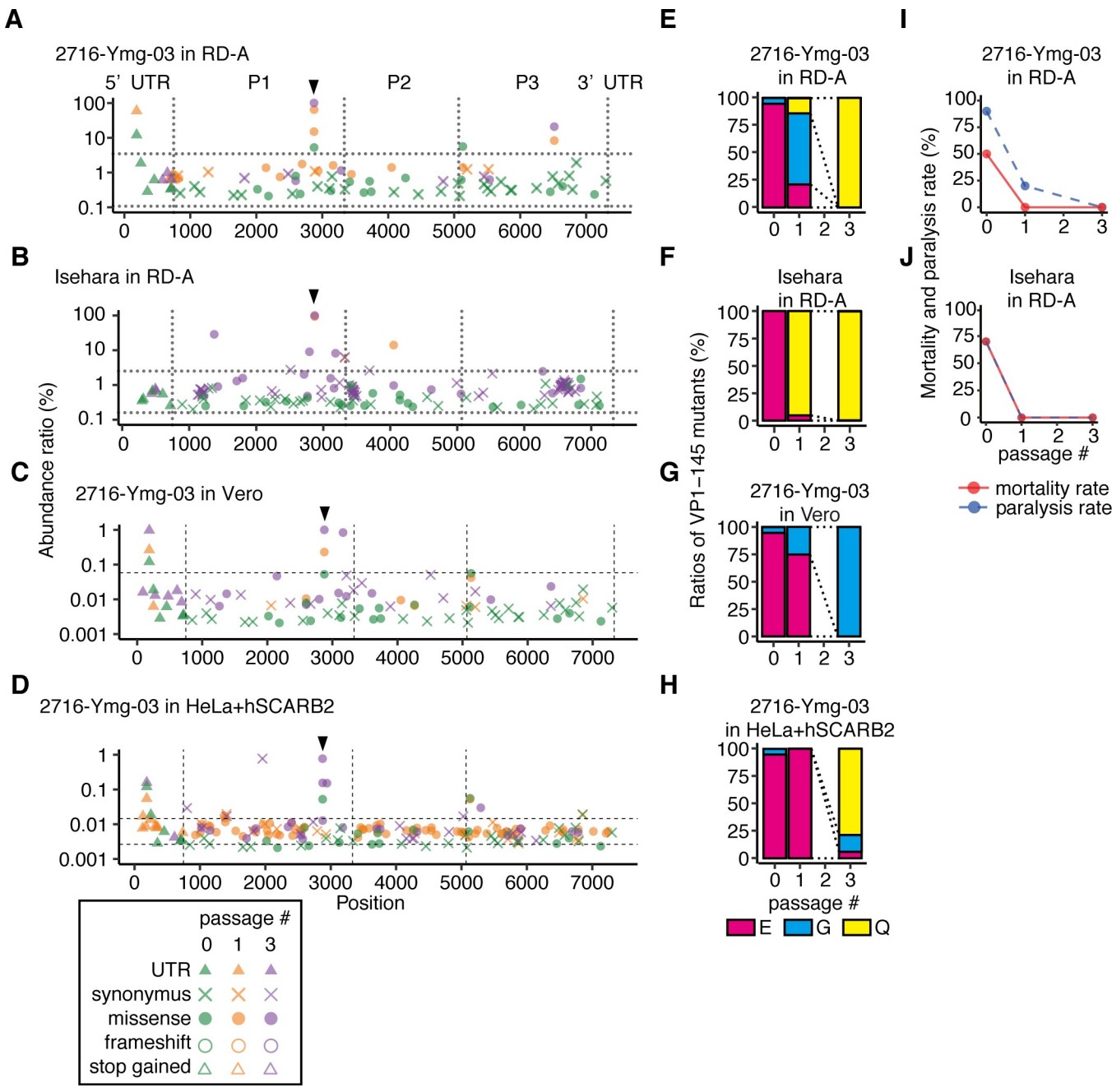

**Fig 1. SNVs and virulence of EV71 populations passaged in RD-A cells.** 2716-Ymg-03 (A, C–E, G–I) and Isehara (B, F, J) strains were passaged in RD-A cells (A–B, E–F, I–J), Vero cells (C, G), and HeLa+hSCARB2 (D, H) for three generations. (A–D) To detect SNVs in the passaged virus populations, NGS reads were mapped to the consensus sequence of the p-0 virus and analyzed using LoFreq and SnpEff software. The abundance of each annotated SNV was plotted on the corresponding position in the virus genome. Vertical dotted lines indicate borders separating genomic regions (5'UTR, P1, P2, P3, and 3'UTR). Horizontal dotted lines indicate the 1.5 interquartile ranges (IQR) of each SNV mapped to the graph. Arrowheads indicate the position corresponding to VP1-145. (E–H) The abundance of VP1-145 mutants in each virus population before and after passage, estimated from NGS data using ShoRAH software. (I–J) The passaged viruses were used to inoculate hSCARB2 tg mice [$10^6$ $TCID_{50}$/mouse for the 2716-Ymg-03 strain, $10^5$ $TCID_{50}$/mouse for the Isehara strain; intraperitoneal (ip) injection]. Paralysis and death of the infected mice were observed over 14 days.

the population. Together, VP1-145G and Q mutations were the main mutations in both the 2716-Ymg-03 and Isehara strains, which were reproducibly enriched during passage in RD-A cells. The results suggest that strong selection pressure acts on this site.

To confirm whether an adaptive mutation at VP1-145 and was selected for in other cells, we passaged the 2716-Ymg-03 strain in Vero and HeLa-S3 cells. Because of the low susceptibility of HeLa-S3 cells to EV71, we used HeLa-S3 overexpressing flag-tagged hSCARB2 (HeLa+-hSCARB2) for EV71 passages. In the 2716-Ymg-03 strain passaged in Vero cells three times, we detected nearly complete displacement by the VP1-145G mutant (Fig 1C–1G). In the case of HeLa+hSCARB2, the VP1-145Q mutant predominated in the virus population after three passages (Fig 1D–1H). The results suggested that the VP1-145 mutation occurs commonly in other cultured cells as well.

## Cell-adapted viruses are attenuated

To evaluate the virulence of EV71 after cell culture adaptation, the parental viruses (p-0) and the passaged viruses (p-1 and p-3) of the 2716-Ymg-03 and Isehara strains in RD-A cells were tested for virulence using hSCARB2 tg mice (Fig 1I and 1J). The parental viruses of the both strains were lethal to a subset of mice under these inoculation conditions (50% and 70%, respectively). However, after just one passage in RD-A cells, neither of the strains were lethal in mice. These data suggest that avirulent mutants were rapidly selected in the cultured cells.

## EV71 replicates in HS-deficient and hSCARB2-overexpressing cells without mutation at VP1-145 or attenuation

Surprisingly, the viral population was almost completely replaced by VP1-145G or Q mutants after only a few passages (Fig 1). These results led us to hypothesize that the infection efficiency of VP1-145E in cultured cells is restricted because of the low surface expression of SCARB2. Once adaptive mutants emerge through genome replication by low fidelity RNA-dependent RNA polymerases, they immediately become dominant in the virus population. In addition to this, we hypothesized that HS expressed on the surface of cultured cells plays an important role in the selection of these mutants. It is already known that VP1-145 mutation affects the binding of EV71 virions to HS and PSGL-1 but does not significantly affect the binding affinity to hSCARB2 [13, 24, 33, 39]. Many types of cells, including RD-A, Vero, and HeLa-S3 cells, express HS on the cell surface (Fig 2A), but PSGL-1 expression is not observed in this cell line [32]. Our hypothesis is that HS-binding mutants expressing G or Q at VP1-145 more efficiently infect cultured cells by binding to HS on the cell surface, resulting in the selection of the VP1-145G and VP1-145Q mutants.

To confirm this possibility, we generated RD-A cells lacking HS by knocking out the *EXT1* or *EXT2* genes (RDΔEXT1 or RDΔEXT2), which are involved in elongation of the HS chain. In the knockout cell lines, no expression of HS on the cell surface was observed (Fig 2B). Then, flag-tagged hSCARB2 was introduced into wild-type RD-A (RD+hSCARB2) and HS-deficient RD-A cells (RDΔEXT1+hSCARB2 and RDΔEXT2+hSCARB2). The expression of flag-tagged hSCARB2 was confirmed by western blotting (Fig 2C). These cells are referred to as ΔEXT1+-hSCARB2 and ΔEXT2+hSCARB2, respectively, for simplicity. The 2716-Ymg-03 strain was passaged in these cells (Fig 3). Selection of VP1-145G and VP1-145Q mutants in the 2716-Ymg-03 virus population passaged in ΔEXT1 and ΔEXT2 cells was observed (Fig 3A, 3B, 3F and 3G), similar to our observations in RD-A cells (Figs 1A and 3C). The VP1-145G mutant was selected in RD+hSCARB2 cells (Fig 3C and 3H). By contrast, no increase in VP1-145G or VP1-145Q mutants was observed when cells were passaged in the ΔEXT1+hSCARB2 and ΔEXT2+hSCARB2 cells (Fig 3D, 3E, 3I and 3J). This result demonstrates that expressions of HS and hSCARB2 are involved in the rapid selection of VP1-145 mutants *in vitro*.

The virulence of the passaged 2716-Ymg-03 virus populations was tested using hSCARB2 tg mice (Fig 3K–3O). Although the 2716-Ymg-03 populations passaged in ΔEXT1, ΔEXT2, or

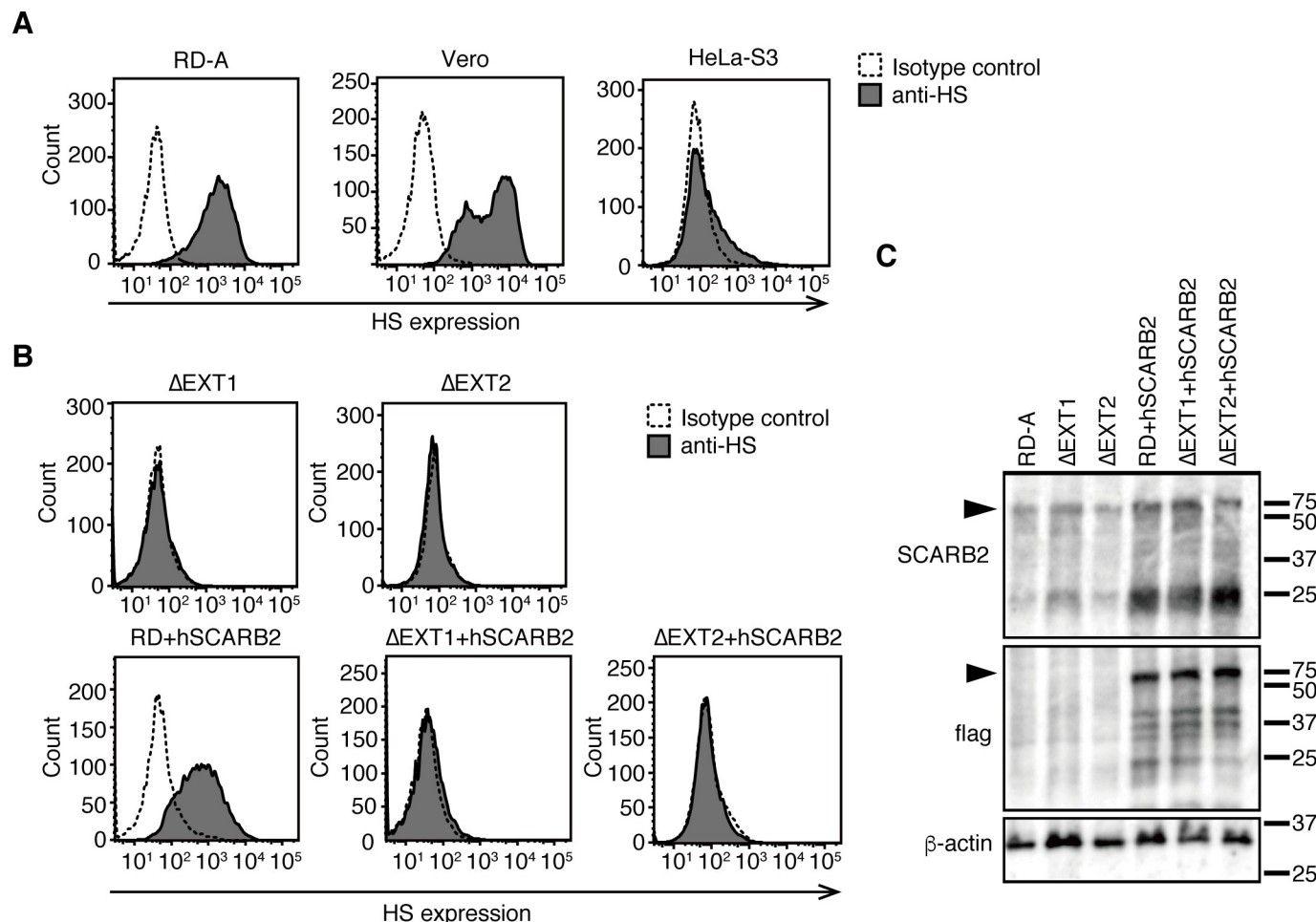

**Fig 2. HS and SCARB2 expression in genetically modified RD-A cells.** (A–B) FACS analysis of HS expression at the cell surface. (C) Western blotting analysis using anti-SCARB2 (top), anti-flag (middle), and anti-ß-actin (bottom) antibodies. The arrowheads indicate hSCARB2.

RD+hSCARB2 cells became avirulent (Fig 3K–3M), similarly to the viruses passaged in RD-A cells (Fig 1E), the virus populations passaged in ΔEXT1+hSCARB2 or ΔEXT2+hSCARB2 cells remained virulent even after three passages (Fig 3N and 3O).

When the Isehara strain was passaged in these cells, the VP1-145Q mutant was selected in ΔEXT1, ΔEXT2, and RD+hSCARB2 cells (Fig 4A–4C and 4F–4H), as was observed in the RD-A cells (Fig 1B and 1D). However, in ΔEXT1+hSCARB2 and ΔEXT2+hSCARB2 cells, VP1-145E remained dominant even after three passages (Fig 4D, 4E, 4I and 4J). The virus populations passaged in ΔEXT1, ΔEXT2, and RD+hSCARB2 cells became avirulent in mice (Fig 4K–4M), but those passaged in ΔEXT1+hSCARB2 and ΔEXT2+hSCARB2 cells remained virulent (Fig 4N and 4O). These results demonstrate that EV71 replicates in HS-deficient and hSCARB2-overexpressing cells without E-to-G or E-to-Q mutation in VP1-145 or attenuation, but that either HS deletion or hSCARB2 overexpression alone is insufficient to avoid mutation.

## VP1-145 mutations occurred and were selected in infectious cDNA-derived virus by passage in RD-A cells

To confirm the hypothesis suggested above, we next investigated how mutations accumulated in the viral genome of a homogenous population. In the experiments described above, we used

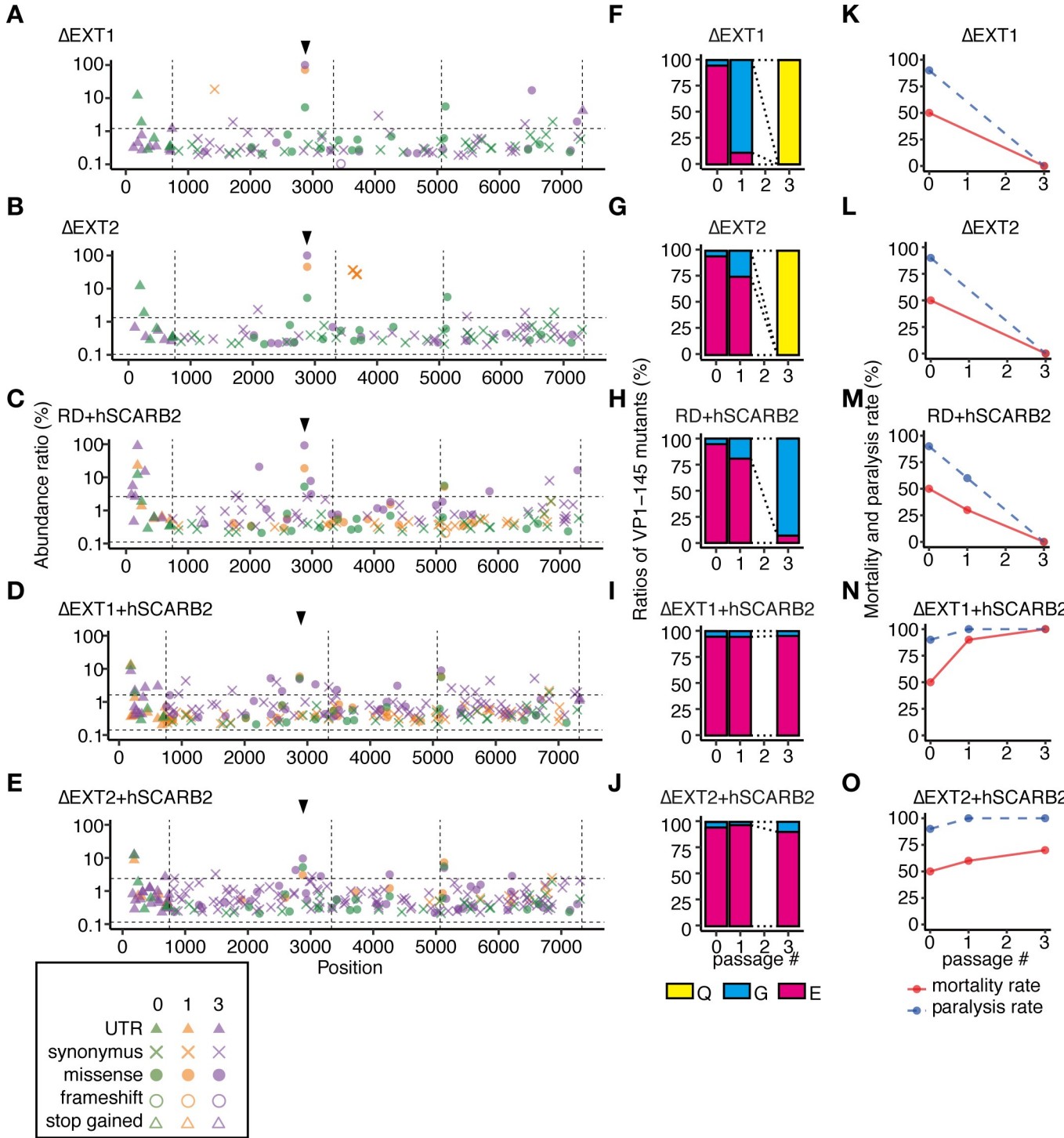

**Fig 3. The 2716-Ymg-03 strain replicates in HS-deficient and hSCARB2-overexpressing cells without attenuation or mutation in VP1-145.** The 2716-Ymg-03 strain was passaged in genetically modified RD-A cells. (A–E) To detect SNVs in passaged virus populations, NGS reads were mapped to the consensus sequence of the p-0 virus and analyzed using LoFreq and SnpEff software. The abundance of each annotated SNV was plotted on the corresponding position in the virus genome. Vertical dotted lines indicate borders separating genetic regions (5'UTR, P1, P2, P3, and 3'UTR). Horizontal dotted lines indicate the 1.5 IQR of the abundance of each SNV mapped to the graph. Arrowheads indicate the position corresponding to VP1-145. (F–J) The abundance of VP1-145 mutants in the virus population before and after passage, estimated from NGS data using ShoRAH software. (K–O) The passaged 2716-Ymg-03 strains were used to infect hSCARB2 tg mice ($10^6$ $TCID_{50}$, ip). Paralysis and death of the infected mice were observed over 14 days.

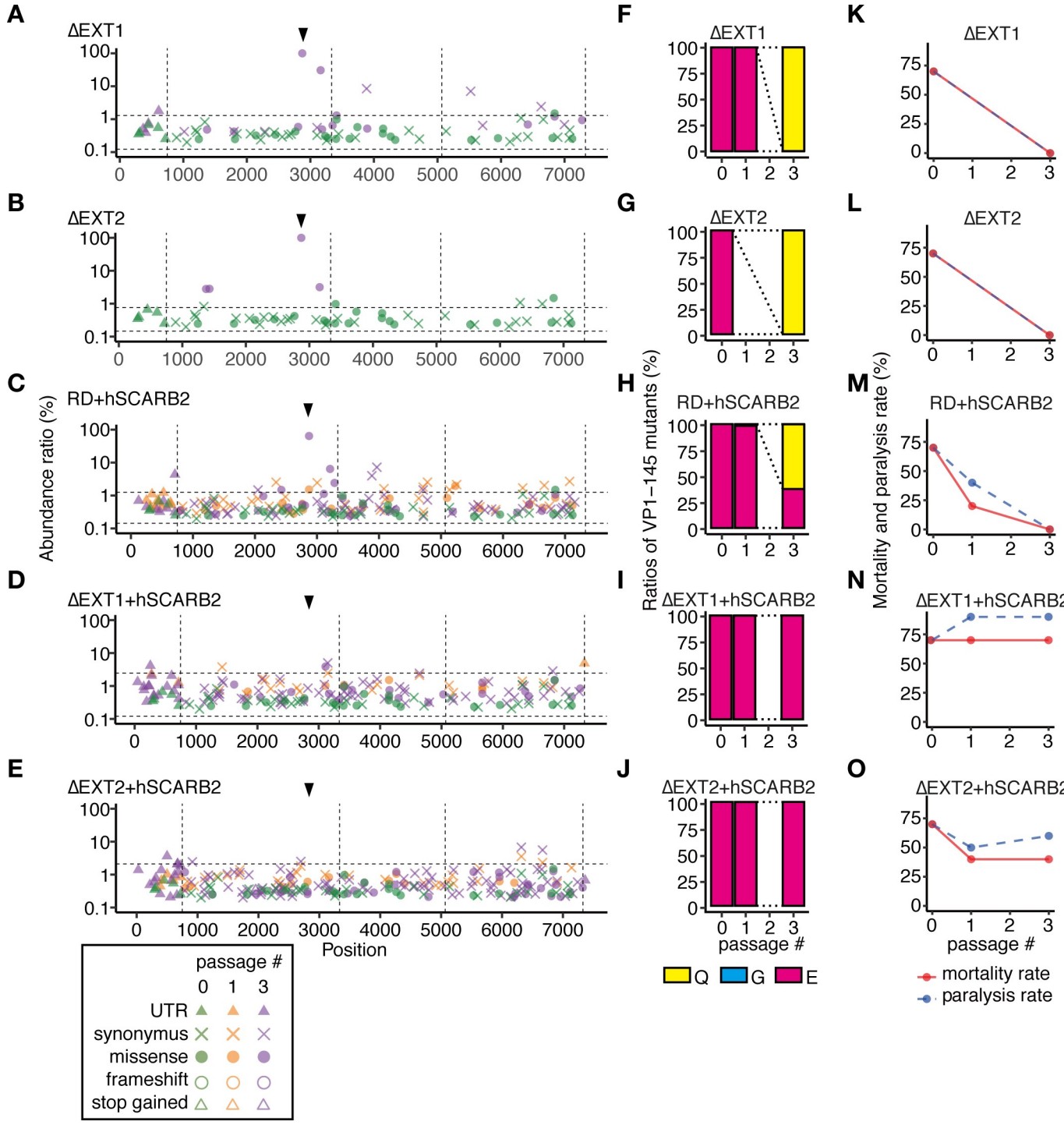

**Fig 4. The Isehara strain replicates in HS-deficient and hSCARB2-overexpressing cells without attenuation or mutation in VP1-145.** The Isehara strain was passaged in genetically modified RD-A cells. (A–E) To detect SNVs in the passaged virus populations, NGS reads were mapped to the consensus sequence of the p-0 virus and analyzed using LoFreq and SnpEff software. The abundance of each annotated SNV was plotted on the corresponding position in the virus genome. Vertical dotted lines indicate borders separating genetic regions (5'UTR, P1, P2, P3, and 3'UTR). Horizontal dotted lines indicate the 1.5 IQR of the abundance of each of the SNVs mapped to the graph. Arrowheads indicate the position corresponding to VP1-145. (F–J) The abundance of VP1-145 mutants in the virus population before and after passage, estimated from NGS data using ShoRAH software. (K–O) Passaged Isehara strains were used to infect hSCARB2 tg mice ($10^5$ TCID$_{50}$, ip). Paralysis and death of the infected mice were observed over 14 days.

p-0 viruses that were not completely homogenous. To minimize the effects caused by the heterogeneity of the starting population, experiments were conducted using *in vitro*-transcribed RNA, which is theoretically quite homogenous. Viral genomic RNA of Isehara strains expressing VP1-145E (Isehara-E), VP1-145G (Isehara-G), or VP1-145Q (Isehara-Q) transcribed from cloned cDNA was transfected into RD-A and ΔEXT1+hSCARB2 cells, and serially passaged three times (Fig 5). For each virus, five independent experiments were performed to assess the reproducibility of selected mutations and the stochastic effect of replication errors in the EV71 genome.

When we prepared Isehara-E p-0 population in RD-A cells, the cells that had probably received viral RNA showed CPE one day after transfection but failed to go completion within seven days in all five independent experiments. As the passage number increased, CPE developed much faster. By contrast, Isehara-G and Isehara-Q showed full CPE two days after transfection to prepare p-0 populations, suggesting that an almost uniform population of VP1-145E virus can hardly infect RD-A cells. When Isehara-E virus was passaged in RD-A cells, VP1-145E virus was diminished by passage in all experiments (Fig 5A). The VP1-145G mutant became dominant in two experiments, while the VP1-145Q mutant became dominant in the other three experiments. In addition, a large number of other nonsynonymous mutations were observed in the P1 region (the area boxed with a red dashed line in Fig 5G). When Isehara-E virus was passaged in ΔEXT1+hSCARB2 cells, CPE appeared within three days after transfection and the following passages, which was much faster than the appearance of CPE in RD-A cells. VP1-145E virus remained dominant in all five experiments (Fig 5B). The abundance of additional nonsynonymous mutations in the P1 region was much lower than in experiments with RD-A cells (the area boxed with a red dashed line in Fig 5H). The result suggested that Isehara-E was unstable in RD-A cells but stable in ΔEXT1+hSCARB2 cells.

When Isehara-G virus was passaged in RD-A cells, the VP1-145G mutation was stable (Fig 5C). In one of five passage experiments with Isehara-G virus using ΔEXT1+hSCARB2 cells, we found an amino acid substitution at VP1-145 from G to E (Fig 5D). In the other four experiments, we found low or undetectable levels of VP1-145E in the virus population, which still predominantly contained the VP1-145G virus. In passage experiments with Isehara-Q virus using RD-A and ΔEXT1+hSCARB2 cells, mutation of VP1-145 was not detected in any experiment (Fig 5E and 5F). In experiments with Isehara-G and Isehara-Q viruses passaged in both cells, nonsynonymous mutations in the P1 region were frequently observed (Fig 5I–5L).

We tested the virulence of these passaged viruses in hSCARB2 tg mice (Fig 5M–5R). Isehara-E viruses passaged in ΔEXT1+hSCARB2 cells, which still had E at VP1-145, showed a virulent phenotype (Fig 5N), whereas those passaged in RD-A cells, which had G or Q at VP1-145, showed an avirulent phenotype (Fig 5M). Almost all the virus populations originating from Isehara-G and Isehara-Q viruses passaged in either cell type induced no symptoms under the same inoculation conditions (Fig 5O–5R). These results support the relationship between the amino acid at VP1-145 and virulence. However, in the fourth passage experiment with Isehara-G virus in ΔEXT1+hSCARB2 cells, although the virus predominantly had E at VP1-145 (93.9%), it was much less virulent (10% paralysis rate and 0% mortality rate) (Fig 5P) than the other VP1-145E virus populations originating from Isehara-E virus (Fig 5N).

## Infection efficiency of VP1-145 mutants in RD-A and ΔEXT1+hSCARB2 cells

Selection of certain mutants may result from the emergence of new mutants and competition between the mutants and the parental virus. To understand why VP1-145G and VP1-145Q viruses were rapidly selected in RD-A cells, we estimated the infection efficiency of VP1-145

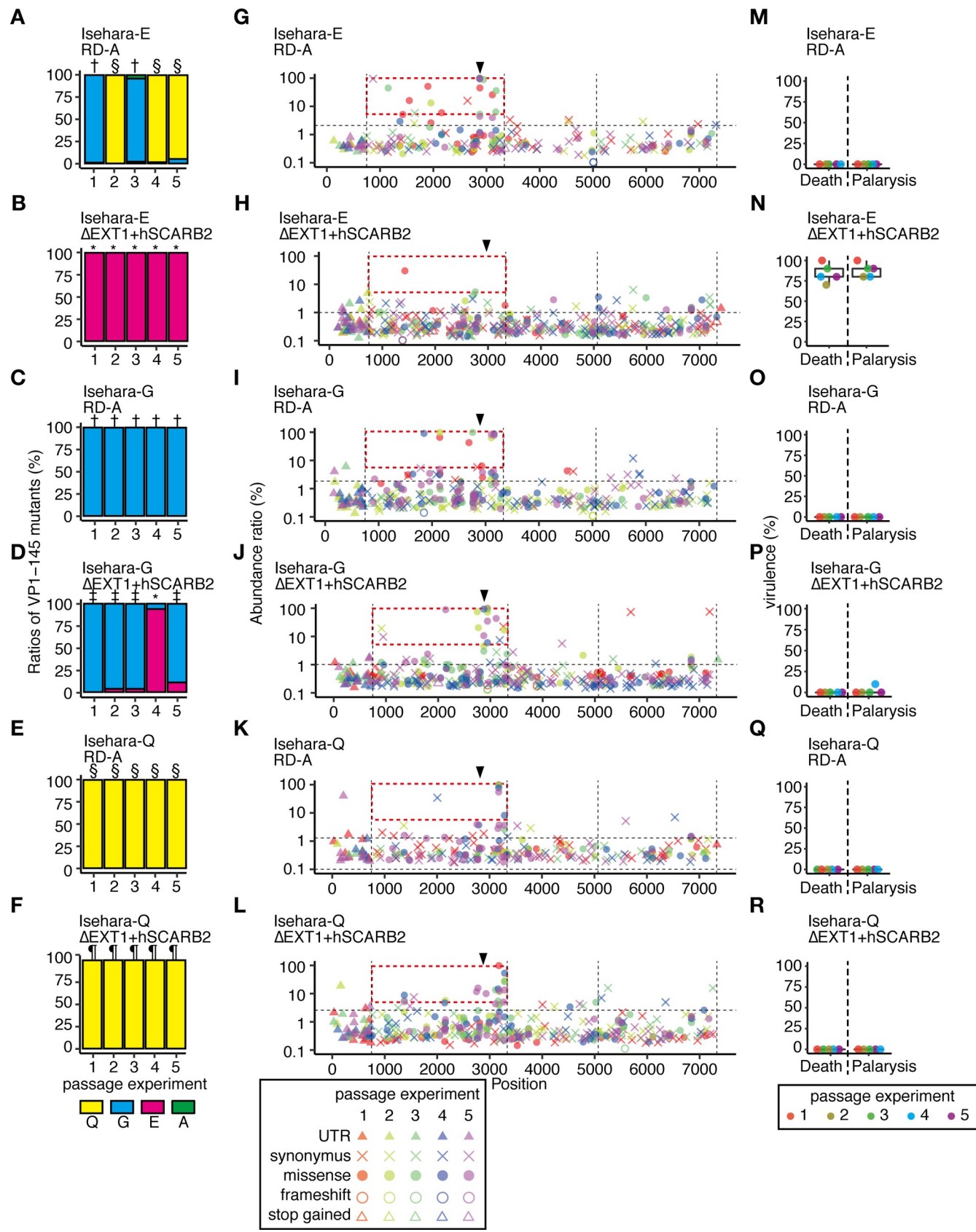

**Fig 5. Genome stability and virulence in EV71 populations with or without VP1-145 mutation that were prepared from cDNA clones.** Viruses recovered from transfection of RD-A or ΔEXT1+hSCARB2 cells with *in vitro*-transcribed RNA from infectious cDNA of Isehara-E, -G, and -Q were passaged in these cells three times. (A–F) The abundance of VP1-145 mutants in the virus population before and after passage, estimated from NGS data using ShoRAH software. VP1-145E-dominant populations resulting from three passages in ΔEXT1+hSCARB2 cells, VP1-145G-dominant populations passaged in RD-A cells, VP1-145G-dominant populations passaged in ΔEXT1+hSCARB2 cells, VP1-145Q-dominant populations passaged in RD-A cells, and VP1-145Q-dominant populations passaged in ΔEXT1+hSCARB2 (analyzed in Figs 6 and 7 and Table 2) are indicated with *, †, ‡, §, and¶, respectively. (G–L) To detect SNVs in passaged virus populations, NGS reads were mapped to the sequence of the cloned cDNA and analyzed using LoFreq and SnpEff software. The abundance of each annotated SNV was plotted on the corresponding position in the virus genome. Vertical dotted lines indicate borders separating genetic regions (5'UTR, P1, P2, P3, and 3'UTR). Horizontal dotted lines indicate the 1.5 IQR of the abundance of each SNV mapped to the graph. Arrowheads indicate the position corresponding to VP1-145. High frequency (>5%) mutations in the P1 region are boxed with a red dashed line. (M–R) Passaged viruses were used to infect hSCARB2 tg mice ($10^5$ TCID$_{50}$, ip). Paralysis and death of the infected mice were observed over 14 days.

mutants. In a viral population with a uniform genome sequence, the infection efficiency ($E$) can be defined using the viral titer ($T$) and the number of viral particles ($P$) in the virus sample using the following equation: $E = T/P$. We determined $T$ and $P$ of the viral populations by microtiter assays and quantitative PCR (qPCR) of the viral RNA genome, and estimated the infection efficiency of the VP1-145E, G, and Q viruses ($E_E$, $E_G$, and $E_Q$, respectively). The virus populations chosen for analysis were the VP1-145E virus passaged in ΔEXT1+hSCARB2 cells (experiment 5), the VP1-145G virus passaged in RD-A cells (experiment 3), and the VP1-145Q virus passaged in RD-A cells (experiment 1), which are shown in Fig 5H–5, 5I–3 and 5K–1, respectively, as representative populations for each virus. $E_G$ [0.005 median tissue culture infectious dose (TCID$_{50}$)/copy] and $E_Q$ (0.01 TCID$_{50}$/copy) were 238 and 489 times higher than $E_E$ in RD-A cells (0.000021 TCID$_{50}$/copy), respectively (Table 1). Thus, once the VP1-145G or Q virus appears in the population during replication, these mutants will infect RD-A cells with more than a 200-fold higher infection efficiency than the parental virus. This result explains, at least in part, the rapid selection of the VP1-145G and Q mutations in RD-A cells. On the other hand, in ΔEXT1+hSCARB2 cells, the infection efficiency of these viruses was notably increased; in particular, $E_E$ increased approximately 100-fold. The relative difference between $E_E$ and $E_G$ or $E_Q$ decreased by14–16-fold. This result indicates that selection pressure on the VP1-145G or VP1-145Q mutant is lower in ΔEXT1+hSCARB2 cells than in RD-A cells. Although these estimates might be affected by mutations at other genomic positions that modulate infection efficiency, they are consistent with the population analysis above. According to our previous report [13], the binding efficiency of VP1-145G virus to heparin-agarose is 100–300-fold higher than that of VP1-145E virus, suggesting that the difference in infection efficiency is dependent on the efficiency of viral binding affinity to HS.

## Secondary mutations were positively selected following E-to-G or E-to-Q mutation at VP1-145

In addition to VP1-145 mutations, we detected nonsynonymous mutations at a high abundance (>5%) that were concentrated in the P1 region (Fig 5G and 5I–5L). These secondary mutations were observed in almost all combinations of viruses and cells except for the Isehara-E virus passaged in ΔEXT1+hSCARB2 cells, where the original amino acid, VP1-145E, was maintained. In contrast to the P1 region, nonsynonymous mutations were not observed in the

**Table 1. Relative infection efficiency of VP1-145 mutants in RD-A and ΔEXT1+hSCARB2 cells.**

|  | WT<br>TCID$_{50}$/copy (relative value) | ΔEXT1+HSCARB2<br>TCID$_{50}$/copy (relative value) |
|---|---|---|
| Isehara-E | 0.000021 (1) | 0.0019 (1) |
| Isehara-G | 0.0050 (238) | 0.031 (16) |
| Isehara-Q | 0.010 (489) | 0.027 (14) |

P2 and P3 regions. These results raised the possibility that once E-to-G or E-to-Q substitutions at VP1-145 occur (spontaneously or artificially by reverse genetics), secondary mutations in the P1 region are introduced and selected. To confirm this possibility, we first classified the p-3 virus populations shown in Fig 5 into three groups according to the VP1-145 amino acid dominant in the p-3 population, and then further classified them according to the cell line used for passage. The populations were classified as VP1-145E-dominant populations passaged in ΔEXT1+hSCARB2 cells, VP1-145G-dominant populations passaged in RD-A cells, VP1-145G-dominant populations passaged in ΔEXT1+hSCARB2 cells, VP1-145Q-dominant populations passaged in RD-A cells, or VP1-145Q-dominant populations passaged in ΔEXT1+hSCARB2 cells, and are indicated by *, †, ‡, §, and ¶ in Fig 5A–5F.

Nonsynonymous mutations detected in at least two independent experiments shown in Fig 5 at more than 5% abundance are shown in Table 2. Many other minor mutations that appeared only once at a low abundance were not subjected to analysis. We noticed some mutations that appeared preferentially in particular combinations of cells and viruses. In VP1-145G-dominant populations passaged in RD-A cells (indicated by † in Fig 5A and 5C), six mutations, at VP3-144, VP1-164, VP1-169, VP1-224, VP1-241, and VP1-246, fit the above criteria. Of these, three mutations, at VP3-144, VP1-224, and VP1-241, were selected in more than two virus populations of this group. Notably, A-to-V mutation at VP1-224 was detected in five passage experiments at a high abundance (26–87%). In VP1-145Q-dominant populations passaged in RD-A cells (indicated by § in Fig 5A and 5E), two secondary mutations were detected (Table 2). One of them, a P-to-A mutation at VP1-246, was observed in all five independent passage experiments at a high abundance (55–99%). To analyze the spatial distribution of these mutations, we mapped the secondary mutations on the pentameric structure of the EV71 capsid (Fig 6). VP1-224 was located surrounding the pocket factor binding site (Fig 6A). On the other hand, VP1-246 protruded from the virus capsid near the 5-fold axis (Fig 6B). Moreover, in VP1-145G-dominant populations in ΔEXT1+hSCARB2 cells (indicated by ‡ in Fig 5D), L-to-F mutation at VP1-169 was preferentially selected in four independent passage experiments at a high abundance (35–99%), which was different from the results seen in RD-A cells. This residue was not spatially adjacent to VP1-224, where the mutation was selected in RD-A cells (Fig 6C). In contrast to the above passage conditions, eight secondary mutations were observed in VP1-145Q-dominant populations passaged in ΔEXT1+hSCARB2 cells (indicated by ¶ in Fig 5F), but the abundance of these mutations was mostly lower than 50%, suggesting relatively low selection pressure under this passage condition. These results suggested that secondary mutations located at spatially distinct positions on the capsid were selected depending on a combination of the VP1-145 amino acid and the expression of HS and SCARB2 on the host cells, probably due to different selection pressures.

To confirm whether the secondary mutations were positively selected, we further analyzed the nucleotide diversity ($\pi$) of select virus populations using the NGS data and SNPGENIE software [40]. The ratios between the nucleotide diversity at nonsynonymous sites ($\pi_N$) and synonymous sites ($\pi_S$) can be used to estimate the balance among neutral selection ($\pi_N/\pi_S = 1$), positive selection ($\pi_N/\pi_S > 1$), and negative selection ($\pi_N/\pi_S < 1$). Positive selection enhances nonsynonymous mutation to select more adaptive variants in an environment. In other words, if the fitness of the virus population is low, variants with nonsynonymous mutations conferring higher fitness will be selected, resulting in an increase in the $\pi_N/\pi_S$ ratio. In Fig 7, the $\pi_N/\pi_S$ ratios in the P1 region of VP1-145G- and VP1-145Q-dominant populations passaged in RD-A and ΔEXT1+hSCARB2 cells were higher than those of VP1-145E-dominant populations. This result is evidence of positive selection in the P1 region in association with VP1-145 mutation. On the other hand, we found no apparent correlation between VP1-145 mutation and $\pi_N/\pi_S$ ratios in the P2 and P3 regions. This evidence supports the induction of

secondary nonsynonymous mutations in the P1 region in association with VP1-145G and Q mutations.

## Mutation of VP1-145 exclusively correlated with EV71 virulence

We further tested the possibility that mutations other than VP1-145, including secondary mutations, contribute to EV71 attenuation. In all the codon positions where SNVs were

**Table 2. List of secondary mutations detected at greater than 5% abundance in the P1 region of VP1-145G- and VP1-145Q-dominant populations.**

| Virus population[a] | Cells | Original virus | Position | Amino acid substitution | Mutation frequency | Structural position |
|---|---|---|---|---|---|---|
| G | Fig 5I–1 | RD-A | Isehara-G | VP3-144 | K to Q | 65 | Outer surface |
| G | Fig 5I–2 | RD-A | Isehara-G | VP3-144 | K to M | 98 | Outer surface |
| G | Fig 5I–1 | RD-A | Isehara-G | VP1-164 | D to E | 6 | Outer surface |
| G | Fig 5G–3 | RD-A | Isehara-E | VP1-169 | L to F | 89 | Outer surface |
| G | Fig 5G–1 | RD-A | Isehara-E | VP1-224 | A to V | 26 | Pocket factor binding site |
| G | Fig 5I–2 | RD-A | Isehara-G | VP1-224 | A to V | 63 | Pocket factor binding site |
| G | Fig 5I–3 | RD-A | Isehara-G | VP1-224 | A to V | 83 | Pocket factor binding site |
| G | Fig 5I–4 | RD-A | Isehara-G | VP1-224 | A to V | 87 | Pocket factor binding site |
| G | Fig 5I–5 | RD-A | Isehara-G | VP1-224 | A to V | 78 | Pocket factor binding site |
| G | Fig 5I–4 | RD-A | Isehara-G | VP1-241 | S to L | 87 | Outer surface |
| G | Fig 5I–5 | RD-A | Isehara-G | VP1-241 | S to L | 78 | Outer surface |
| G | Fig 5G–3 | RD-A | Isehara-E | VP1-246 | P to A | 6 | Outer surface |
| Q | Fig 5K–1 | RD-A | Isehara-Q | VP1-246 | P to A | 99 | Outer surface |
| Q | Fig 5K–2 | RD-A | Isehara-Q | VP1-246 | P to A | 96 | Outer surface |
| Q | Fig 5K–3 | RD-A | Isehara-Q | VP1-246 | P to A | 96 | Outer surface |
| Q | Fig 5K–4 | RD-A | Isehara-Q | VP1-246 | P to A | 77 | Outer surface |
| Q | Fig 5K–5 | RD-A | Isehara-Q | VP1-246 | P to A | 55 | Outer surface |
| Q | Fig 5K–4 | RD-A | Isehara-Q | VP1-282 | N to D | 8 | Outer surface |
| G | Fig 5J–2 | ΔEXT1+hSCARB2 | Isehara-G | VP1-104 | N to D | 75 | Outer surface |
| G | Fig 5J–1 | ΔEXT1+hSCARB2 | Isehara-G | VP1-169 | L to F | 99 | Outer surface |
| G | Fig 5J–2 | ΔEXT1+hSCARB2 | Isehara-G | VP1-169 | L to F | 73 | Outer surface |
| G | Fig 5J–3 | ΔEXT1+hSCARB2 | Isehara-G | VP1-169 | L to F | 94 | Outer surface |
| G | Fig 5J–5 | ΔEXT1+hSCARB2 | Isehara-G | VP1-169 | L to F | 35 | Outer surface |
| G | Fig 5J–5 | ΔEXT1+hSCARB2 | Isehara-G | VP1-224 | A to V | 44 | Pocket factor binding site |
| Q | Fig 5L–3 | ΔEXT1+hSCARB2 | Isehara-Q | VP2-141 | T to M | 6 | Disordered region |
| Q | Fig 5L–4 | ΔEXT1+hSCARB2 | Isehara-Q | VP2-141 | T to M | 9 | Disordered region |
| Q | Fig 5L–5 | ΔEXT1+hSCARB2 | Isehara-Q | VP2-141 | T to M | 5 | Disordered region |
| Q | Fig 5L–5 | ΔEXT1+hSCARB2 | Isehara-Q | VP1-104 | N to D | 12 | Outer surface |
| Q | Fig 5L–5 | ΔEXT1+hSCARB2 | Isehara-Q | VP1-164 | D to E | 10 | Outer surface |
| Q | Fig 5L–5 | ΔEXT1+hSCARB2 | Isehara-Q | VP1-224 | A to V | 13 | Outer surface |
| Q | Fig 5L–2 | ΔEXT1+hSCARB2 | Isehara-Q | VP1-241 | S to L | 5 | Outer surface |
| Q | Fig 5L–3 | ΔEXT1+hSCARB2 | Isehara-Q | VP1-241 | S to L | 15 | Outer surface |
| Q | Fig 5L–1 | ΔEXT1+hSCARB2 | Isehara-Q | VP1-246 | P to A | 98 | Outer surface |
| Q | Fig 5L–2 | ΔEXT1+hSCARB2 | Isehara-Q | VP1-249 | I to V | 12 | Buried |
| Q | Fig 5L–5 | ΔEXT1+hSCARB2 | Isehara-Q | VP1-249 | I to V | 12 | Buried |
| Q | Fig 5L–2 | ΔEXT1+hSCARB2 | Isehara-Q | VP1-282 | N to D | 29 | Outer surface |
| Q | Fig 5L–3 | ΔEXT1+hSCARB2 | Isehara-Q | VP1-282 | N to D | 26 | Outer surface |
| Q | Fig 5L–4 | ΔEXT1+hSCARB2 | Isehara-Q | VP1-282 | N to D | 53 | Outer surface |
| Q | Fig 5L–5 | ΔEXT1+hSCARB2 | Isehara-Q | VP1-282 | N to D | 13 | Outer surface |

[a] The left column shows the amino acid residue in VP1-145 predominantly detected in each virus population, and the right column shows the corresponding figure.

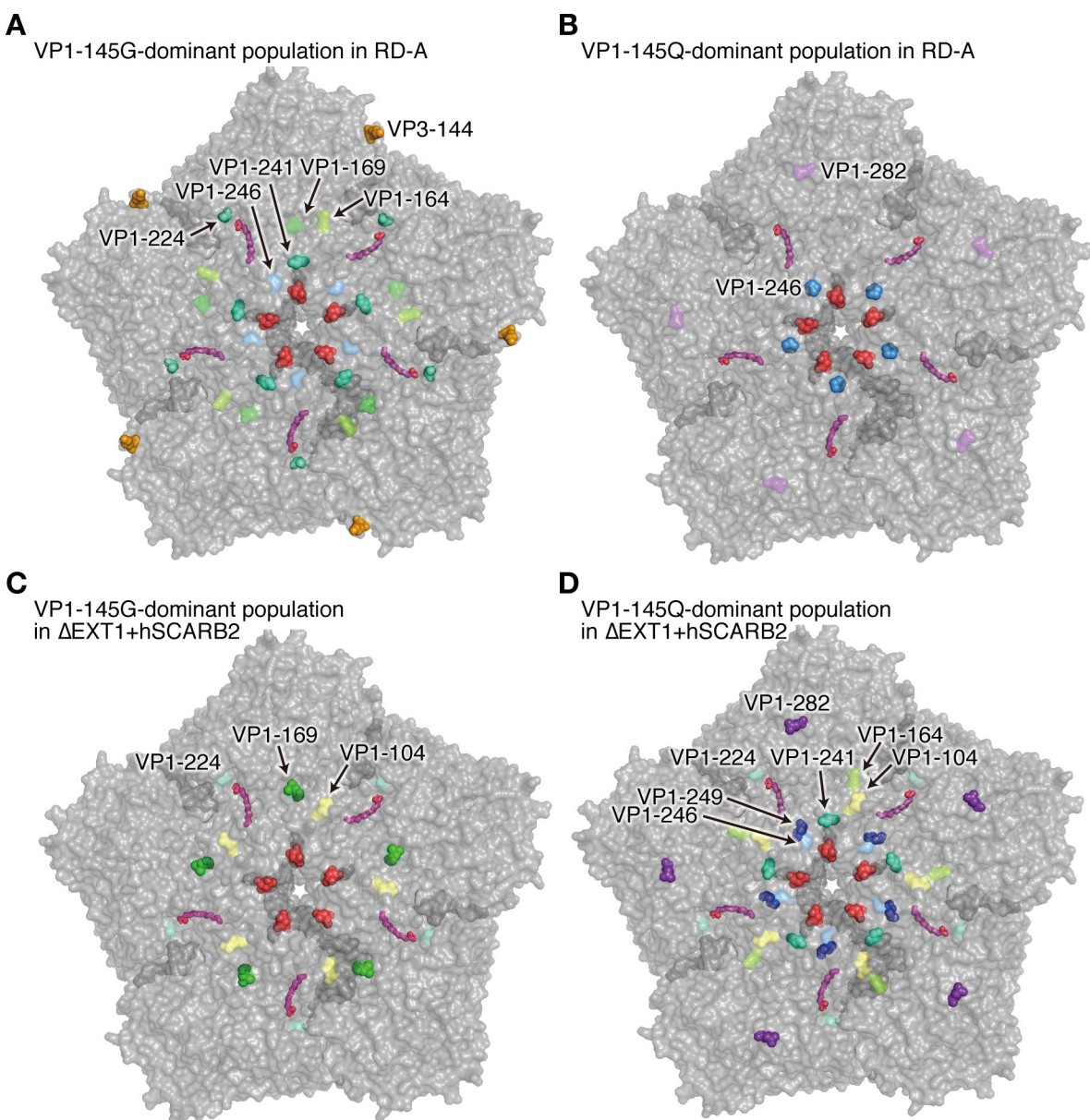

**Fig 6. Distribution of secondary mutations induced in VP1-145G and Q mutants.** Translucent surface representation of the pentameric structure of the EV71 capsid (Protein Data Bank: 4AED) viewed along the 5-fold axis from the outside. The amino acid residue VP1-145 is indicated in red. The amino acid residues of secondary mutations detected in VP1-145G-dominant populations passaged in RD-A cells (A), VP1-145Q-dominant populations passaged in RD-A cells (B), VP1-145G-dominant populations passaged in ΔEXT1+hSCARB2 cells (C), and VP1-145Q-dominant populations passaged in ΔEXT1+hSCARB2 cells (D), which are marked in Fig 5A–5F with †, ‡, §, and¶, respectively. Residues in which secondary mutations were detected in at least two virus populations in each group are represented by spheres. Pocket factors are indicated in magenta.

detected (Fig 5G–5L), the Spearman's correlation coefficient (rho) between the mutation rate and the rate of paralysis or mortality was determined and plotted their significance [-log$_{10}$ (*p*-value)] at the corresponding codon position (Fig 8). Since mutations increasing in a virus population transitioning from a virulent to an avirulent phenotype should negatively correlate with virulence (rho < 0), we focused on negatively correlated mutations. We found that mutation at VP1-145 exclusively showed a significant negative correlation with EV71 virulence.

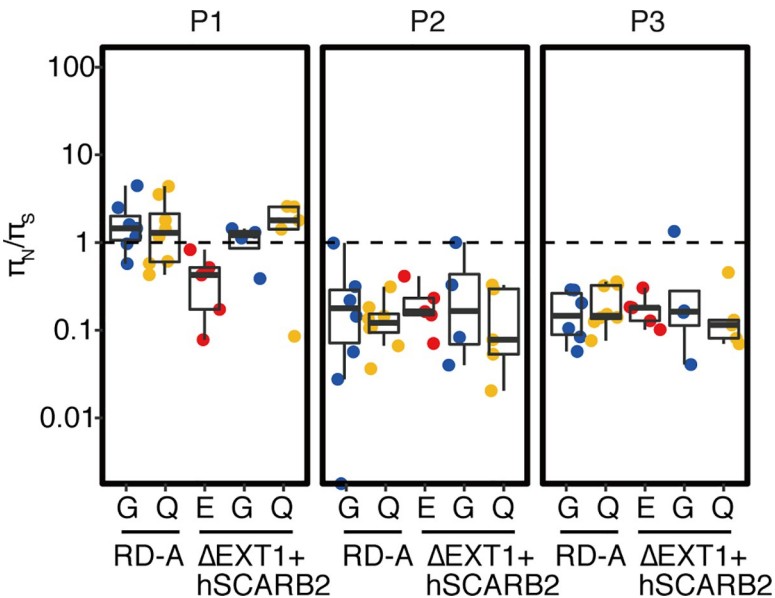

**Fig 7. VP1-145 mutation promotes secondary mutations in the P1 region.** The virus populations shown in Fig 5 that predominantly (greater than 50%) contained VP1-145E, VP1-145G, or VP1-145Q mutants were further divided into two groups according to whether they arose from passage in RD-A cells or ΔEXT1+hSCARB2 cells. In each genetic region (P1, P2, and P3), the nucleotide diversity of synonymous and nonsynonymous sites ($\pi_S$ and $\pi_N$, respectively) was calculated using SNPgenie software. $\pi_N/\pi_S$ ratios are illustrated by scatter and box plots.

## Discussion

EV71 freshly isolated from clinical samples is difficult to grow in cell culture. To overcome this, the virus employs a strategy that allows it to survive under the adverse conditions of cell culture. In this study, we clarified the molecular mechanism involved in virus adaptation to cell culture. HS attachment receptor selected HS-binding viruses very rapidly during cell culture adaptation of EV71. Virus populations in clinical samples directly sequenced without propagation in cell culture are mostly composed of VP1-145E viruses [16, 17]. However, after 1–3 passages in wild-type, HS-deficient, or hSCARB2-overexpressing RD-A cells, VP1-145G or VP1-145Q mutants became dominant in all the populations that we tested. These results suggest that the fitness of the virulent EV71 population is low in the cell culture environment probably because of low surface expression of SCARB2 in cultured cells. Once more adaptive mutants with G or Q at VP1-145 arise through the error-prone RNA replication process, these mutants selectively replicate in cultured cells. Our data indicated that these two mutants were occurred by chance (Fig 5A) and we speculated that a mutant first emerging in the population should be selected as a dominant mutation. VP1-145G or Q mutation eventually becomes fixed in the population, resulting in increased fitness. VP1-145G or VP1-145Q viruses can infect RD-A cells with 200–500-fold higher efficiency than VP1-145E viruses (Table 1). This selection is apparently associated with the balance between the expression of HS and hSCARB2 in cultured cells because this selection was not observed in HS-deficient hSCARB2-overexpressing RD-A cells (Figs 3–5). We therefore propose that the major selection pressure during propagation of EV71 in cultured cells is the difference in infection efficiency among the VP1-145 mutants due to their different binding affinities to the HS attachment receptor.

During fixation from VP1-145E to VP1-145G or Q, the diversity of the viral population becomes theoretically low due to selective sweeping. However, we found a variety of nonsynonymous mutations in the P1 region that arose after or in parallel with the VP1-145 mutation

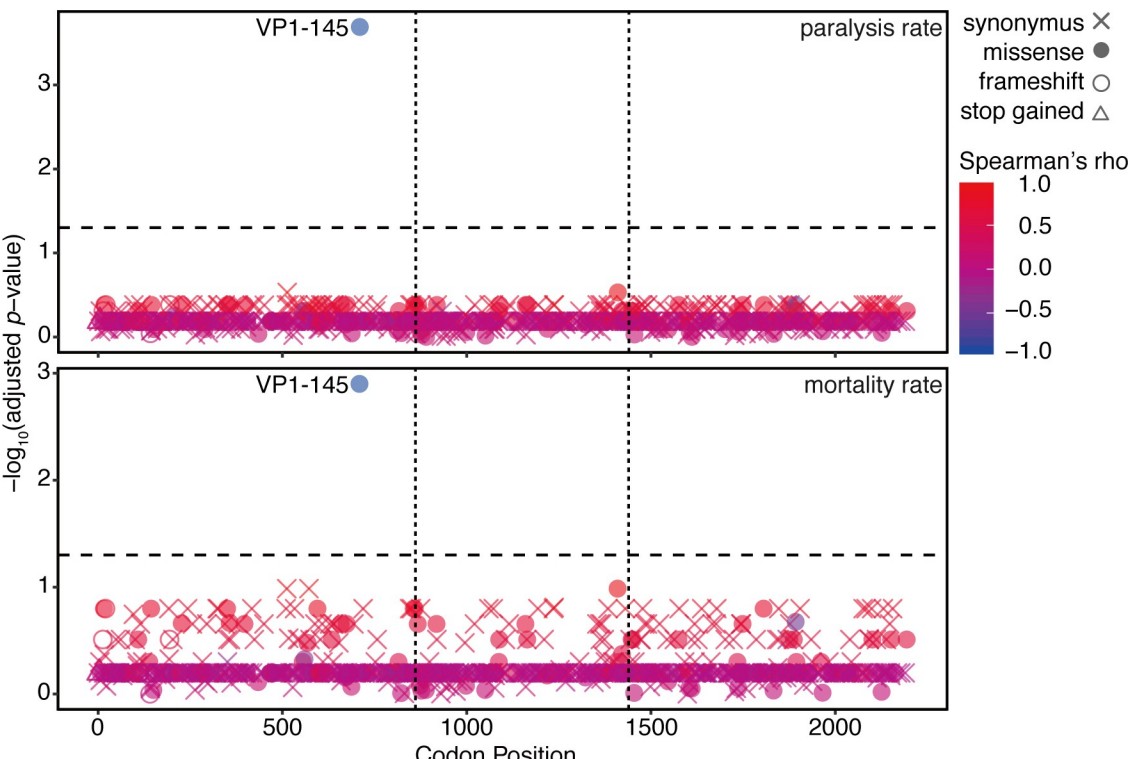

**Fig 8. Correlation between SNVs and EV71 virulence.** SNVs were detected by mapping of the viral reads acquired in the passage experiments in Fig 5 to the consensus sequence of Isehara-E. Spearman's correlation coefficients between the SNVs and the paralysis or mortality rate at all codon positions where SNVs were detected were calculated. Negative log-transformed $p$-values, which were calculated with the null hypothesis that there was no correlation between two factors, were plotted on the corresponding codon positions. Positive and negative correlations in red or blue, respectively. Horizontal dotted lines indicate $p = 0.05$.

(Fig 5G). These secondary mutations were also observed in the populations of VP1-145G and Q virus artificially generated by reverse genetics (Fig 5I–5L). Although the actual triggers for secondary mutations are unknown, we propose some possibilities. Some secondary mutations are beneficial for survival in cultured cells by enhancing affinity to receptor(s), altering receptor specificity, enhancing uncoating, stabilizing capsid conformation, and so on. Regardless of the mechanism, these mutations should increase the fitness of viruses expressing VP1-145G or Q in the cultured cells. In summary, the fitness of the virus during cell adaptation progresses through three stages (Fig 9). The VP1-145E virus has the lowest level of fitness (stage 1). After E-to-G or E-to-Q mutation at VP1-145, the mutant viruses have moderate levels of fitness (stage 2). Finally, viruses that acquire secondary mutations will have the highest levels of fitness (stage 3).

Several previous studies revealed that VP1-145G and Q mutants replicate in cultured cells more rapidly than the VP1-145E virus by interacting with HS [13, 24]. However, deletion of HS is insufficient to prevent the selection of VP1-145G and Q (Figs 3 and 4), indicating that additional host factor(s) contributing to this selection still remain in HS-deficient cells. If the binding of VP1-145G and Q virus to HS is mediated mainly by electrostatic interactions, other negatively charged molecules expressed on the host cell surface may bind to these mutants and support their infection.

Our study demonstrates that attenuation of EV71 during cell culture adaptation is exclusively associated with the VP1-145 mutation (Fig 8), indicating that VP1-145E is adaptive in an *in vivo* environment but deleterious in an *in vitro* environment (stage 1 in Fig 9). Our

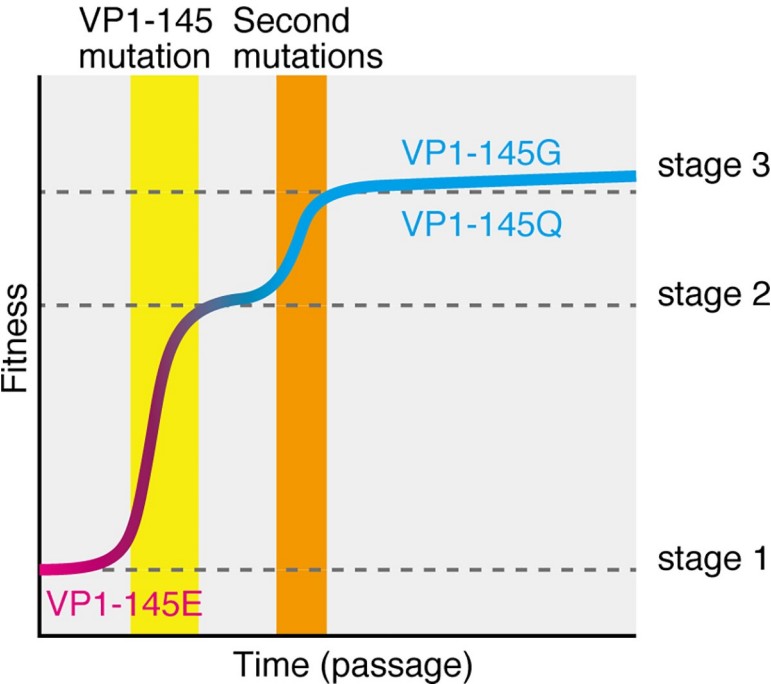

**Fig 9. A model of the fitness landscape of virulent EV71 populations during cell culture adaptation in RD-A cells.**

previous study revealed that VP1-145G is avirulent in hSCARB2 tg mice and cynomolgus monkeys [13, 14]. One possible explanation for these observations is that, because the expression patterns of HS and hSCARB2 are quite different, VP1-145G and Q viruses are adsorbed onto cells with little or no hSCARB2 expression or extracellular matrix via HS, resulting in abortive infection and inefficient dissemination *in vivo*. We also found that the VP1-145G virus was more easily neutralized with neutralizing antibodies. These two mechanisms likely account for the negative selection of VP1-145G and Q viruses *in vivo*. Taken together with the findings of this study, these data suggest that HS is a major factor for positive selection of VP1-145G and Q mutants *in vitro* but for negative selection of these mutants *in vivo*. These opposing selection processes give rise to different virus populations *in vivo* and *in vitro*. This also implies that an unknown attachment receptor(s) specific for VP1-145E virus is expressed in EV71 target cells, such as neuronal cells, but is not expressed in cultured cells, such as RD-A cells.

Similar selections mediated by glycosaminoglycan including HS by cell culture adaptation have been reported in the viruses belonging to the *Flaviviridae* (Japanese encephalitis virus, Murray Valley encephalitis virus, West Nile virus, and Dengue virus) [41–44], *Togaviridae* (Sindbis virus, Venezuelan equine encephalitis virus, Tick-borne encephalitis virus, and Chikungunya virus) [45–48], and *Picornaviridae* (human Rhinovirus (HRV) C15, HRV89, and foot-and-mouth disease virus) [49–51]. These selected viruses were attenuated *in vivo*. It is probable that the similar mechanisms of selection and attenuation observed in EV71 also occur in these viruses and that this principle is applicable in a wide range of viral infectious diseases.

Attenuation of viruses after tissue culture adaptation is known to contribute greatly to the development of some live attenuated vaccines, including poliovirus, measles virus, and yellow fever virus [52]. This study provides an example of an attenuation mechanism of viruses in tissue culture using EV71. Tsai *et al*. [53] used this mutation in a live attenuated EV71 vaccine

candidate together with a mutation that increased the fidelity of the RNA-dependent RNA polymerase. This neurovirulence determinant may be useful for the development of live attenuated EV71 vaccines if the appearance of the revertant can be completely prevented.

ΔEXT1+hSCARB2 cells, in which HS was depleted and hSCARB2 was overexpressed, failed to support VP1-145G/Q selection. This newly developed cell line allows us to isolate and propagate EV71 strains circulating in human populations with minimal levels of mutation. This cell line will therefore be a useful tool for the identification of neurovirulence determinants of EV71 and for preparation of the challenge virus for efficacy tests of vaccines or antiviral drugs.

## Materials and methods

### Ethics statements

Experiments using recombinant DNA and pathogens were approved by the Committee for Experiments using Recombinant DNA and Pathogens at the Tokyo Metropolitan Institute of Medical Science. Experiments using mice were approved by the Animal Use and Care Committee and performed in accordance with the Guidelines for the Care and Use of Animals (Tokyo Metropolitan Institute of Medical Science, 2011, approval number: 17036, 18047, and 19063). In the animal infection experiments, mice with severe paralysis in limbs or more than 30% body weight loss were sacrificed by overdose of isoflurane or cervical dislocation.

### Preparation of cell lines

The cells were propagated in Dulbecco's modified Eagle's medium (DMEM) containing 5% fetal calf serum (FCS). Human RD-A cells and African green monkey Vero cells were kindly provided by Dr. Hiroyuki Shimizu, National Institute of Infectious Diseases, Japan. In addition to these two cell lines, a suspension culture-adapted HeLa cell line (HeLa-S3) [54] was used for the passage of EV71 strains. ΔEXT1, ΔEXT2, and RD+hSCARB2 cells were described previously [13]. ΔEXT1+hSCARB2 and ΔEXT2+hSCARB2 cells were established using retrovirus vectors as described previously [13].

### Viruses

The 2716-Ymg-03 strain was isolated as described previously [16]. Isolated 2716-Ymg-03 was propagated once using RD+hSCARB2 cells. Infectious cDNA from the Isehara-E and Isehara-G strains was described previously [13]. Infectious cDNA from the Isehara-Q strain was established using a previously described method [13]. Briefly, the E-to-Q mutation at amino acid position VP1-145 of the Isehara strain was introduced by amplifying the fragments using a pair of mutated primers (Ise VP1-145Q-2-F: 5'-CTA CCG GGC AAG TTG TCC CGC AAT T-3', Ise VP1-145Q-2-R: 5'-GGG ACA ACT TGC CCG GTA GGC GTG C-3') and the pSVA-EV71-Isehara plasmid as a template. The DNA fragment of the original plasmid was replaced with the corresponding mutated fragment. The parental Isehara strain used in Figs 1 and 4 was recovered by transfection of RD+hSCARB2 cells with RNA that had been transcribed *in vitro* from Isehara-E cDNA using a previously described method [13]. The recovered virus was propagated by two passages in RD+hSCARB2 cells. Parental stocks of the 2716-Ymg-03 and Isehara strains (p-0) were passaged in RD-A cells with or without genetic modifications at a low MOI (<0.01). Infected cells were cultured until approximately more than 80% of infected cells had died. For the passage experiment starting from infectious cDNA, *in vitro*-transcribed RNAs were transfected into each cell line. Recovered viruses were then passaged in the corresponding cell line. The virus titer was determined by microplate

assay using RD-A cells, as previously described [13], and was represented as the $TCID_{50}$ using the Kaeber method [55].

## Animal experiments

Six-to-seven-week-old hSCARB2 tg mice [30] were used for infection experiments. After anesthetization by inhalation of isoflurane, a total of 500 μL of virus solution was inoculated intraperitoneally. A total of ten mice were used per group. The mice were monitored for body weight changes and clinical signs of disease for 2 weeks.

## Flowcytometric analysis

Cells were detached by treatment with PBS containing 0.02% ethylenediaminetetraacetic acid (EDTA). Suspended cells were stained with a mouse anti-HS monoclonal IgM antibody (1:500 dilution; F58-10E4; Amsbio) or an isotype control IgM (1:500 dilution; MM-30; BioLegend) for 30 min on ice and then stained with a Cy3-conjugated anti-IgM antibody (1:200 dilution; Jackson ImmunoResearch) for 30 min on ice in the presence of propidium iodide (PI; 1:100 dilution). Stained cells were analyzed on a LSRFortessa X-20 (BD Biosciences).

## Western blotting

Cells were lysed in RIPA buffer [25 mM Tris (pH 7.5), 150 mM NaCl, 1% NP-40, 1% sodium deoxycholate, 0.1% sodium dodecyl sulfate (SDS), 1 mM EDTA, and protease inhibitor cocktail (Roche)]. Solubilized proteins were denatured in 2× SDS sample buffer at 90˚C for 5 min. Samples were subjected to SDS-polyacrylamide gel electrophoresis (PAGE) and transferred to polyvinylidene fluoride (PVDF) membranes. Proteins were detected using an anti-SCARB2 antibody (AF1966; R&D Systems), an anti-flag antibody (F7425; Sigma-Aldrich), and an anti-β-actin antibody (AC74; Sigma-Aldrich).

## Quantitative PCR of the virus genome

Viral RNA was extracted using the QIAamp Viral RNA Mini Kit (Qiagen) followed by reverse transcription using PrimeScript RT Master Mix (Takara). Quantitative PCR was conducted using enterovirus universal primer pairs [56], SYBR Select Master Mix (Applied Biosystems/ThermoFisher Scientific), and a LightCycler 480-II (Roche Life Science).

## NGS analysis

RNA was extracted from virus preparations using the QIAamp Viral RNA Extraction Mini Kit (Qiagen). The NEBNext RNA Library Prep Kit for Illumina and the NEBNext Multiplex Oligos for Illumina Dual Index Primer Set 1 (both from New England Biosciences) were used for all library preparation. The average fragment length for the libraries was 300 bp. The quality of the libraries was evaluated on an Agilent 2100 Bioanalyzer (Agilent) using the Agilent DNA 1000 Kit (Agilent) and quantitated using the Kapa Library Quantification Kit (Kapa Biosystems). The libraries were sequenced on a MiSeq instrument (Illumina) using a 150 bp paired-end kit.

Adaptor and low quality sequences were trimmed from raw sequence reads using PRINSEQ software version 0.20.4 (http://prinseq.sourceforge.net/index.html) with the following settings: derep, 14; min_len, 70; min_qual_mean, 25; ns_max_n, 1; lc_threshold, 60; trim_tail_left, 5; trim_tail_right, 5; trim_qual_left, 20; trim_qual_right, 20; trim_qual_window, 5. To remove reads derived from host cells, the remaining reads were mapped to the hg38 human reference genome using Bowtie 2 software version 2.3.4.1 with the default settings [57].

Consensus viral genomic sequences were assembled from the hg38 unmapped reads using IVA software version 1.0.8 with the default settings [58]. Contigs generated by the assembly contained human sequences, indicating the presence of human-derived reads in the viral reads. To diminish the effect of human-derived reads on SNV analysis, non-viral contigs were combined with the reference sequence of the 2716-Ymg-03 strain (GenBank: LC375766.1) or the Isehara strain (LC375764.1). The remaining reads were mapped to the combined multi-FASTA file using Smalt software version 0.7.6 (https://www.sanger.ac.uk/science/tools/smalt-0) and converted to BAM files using Samtools software version 1.9-4-gaelf9d8. Generated BAM files were used for variant calling using Lofreq software version 2.1.2 [59]. SVNs were annotated using SnpEff software version 4.3t [60]. To ensure the accuracy of SNV data, SNVs that met the following criteria were discarded: the abundance ratio was less than $10^{-3}$ and the coverage at the position was less than $\log(1-p)/\log(1-f)-1$, where the mutation frequency $f$ is detected with a probability $p$ or better [61]. In this study, $p$ was set to 0.95.

Ratios of mutated codons at VP1-145 were calculated based on the abundance of haplotypes estimated using the shorah.py program of ShoRAH software version 1.1.2 [62] with the following settings: analyzed region, 2800–2920; window size, 120; shift, 3. The estimated abundance of the reconstructed haplotypes and their sequences described in the resulting .popl file were used for calculation of the VP1-145 codon abundance.

The synonymous and nonsynonymous nucleotide diversity of each gene was calculated using SNPGENIE software [40].

## Statistical analysis

Spearman's correlation tests (Fig 8) were performed using R version 3.5.1. In multiple comparisons tests, $p$-values were adjusted by false discovery rate correction (Fig 8).

## Acknowledgments

The authors thank Ms. Ayako Ohkubo for technical assistance and Drs. Tomofumi Nakamura (The Research Foundation for Microbial Diseases of Osaka University) and Yutaka Kano (Genome Dynamics project, Tokyo Metropolitan Institute of Medical Science) for helping with the NGS analysis.

## Author Contributions

**Conceptualization:** Satoshi Koike.

**Data curation:** Katsumi Mizuta.

**Formal analysis:** Kyousuke Kobayashi.

**Funding acquisition:** Satoshi Koike.

**Investigation:** Kyousuke Kobayashi.

**Methodology:** Kyousuke Kobayashi.

**Project administration:** Kyousuke Kobayashi.

**Resources:** Katsumi Mizuta.

**Supervision:** Satoshi Koike.

**Validation:** Kyousuke Kobayashi.

**Visualization:** Kyousuke Kobayashi.

**Writing – original draft:** Kyousuke Kobayashi.

**Writing – review & editing:** Satoshi Koike.

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
