## [Decision Letter · Decision Letter 0]

9 Dec 2019

Dear Dr. Koike:

Thank you very much for submitting your manuscript "Heparan sulfate attachment receptor is a major selection factor for attenuated enterovirus 71 mutants during cell culture adaptation" (PPATHOGENS-D-19-01860) for review by PLOS Pathogens. Your manuscript was fully evaluated at the editorial level and by independent peer reviewers. The reviewers appreciated the attention to an important topic but identified some aspects of the manuscript that should be improved.

We therefore ask you to modify the manuscript according to the review recommendations before we can consider your manuscript for acceptance. Your revisions should address the specific points made by each reviewer.

(1) A letter containing a detailed list of your responses to the review comments and a description of the changes you have made in the manuscript. Please note while forming your response, if your article is accepted, you may have the opportunity to make the peer review history publicly available. The record will include editor decision letters (with reviews) and your responses to reviewer comments. If eligible, we will contact you to opt in or out.

(2) Two versions of the manuscript: one with either highlights or tracked changes denoting where the text has been changed; the other a clean version (uploaded as the manuscript file).

We hope to receive your revised manuscript within 60 days or less. If you anticipate any delay in its return, we ask that you let us know the expected resubmission date by replying to this email.

[LINK]

Sincerely,

Richard J. Kuhn, PhD

Associate Editor

PLOS Pathogens

Raul Andino

Section Editor

PLOS Pathogens

Kasturi Haldar

Editor-in-Chief

PLOS Pathogens

orcid.org/0000-0001-5065-158X

Grant McFadden

Editor-in-Chief

PLOS Pathogens

orcid.org/0000-0002-2556-3526

The reviewers were supportive of this manuscript but had several minor but important comments that should be addressed in a revised version.

Reviewer's Responses to Questions

**Part I - Summary**

Reviewer #1: This manuscript by Kobayashi et al details the fitness of the “virulent” EV71 is low in the cell culture environment. Two major adaptive mutants located in VP1-145G or VP1-145Q were produced through the error-prone RNA replication process and can replicate in the cultured cells and eventually become fixed in the population. Interestingly, both of the cell cultured adaptive mutants are attenuated virus when they were tested for virulence using hSCARB2 mice, indicating that avirulent mutants were rapidly selected in the cultured cells, but not survival in the nature host. Although the authors only used the RD cells for the selection of VP1-145 mutants, the author clearly demonstrated the attenuation of EV-A71 during the cell culture adaptation is exclusively associated with VP1-145 mutation. Scientifically speaking, the design of the entire manuscript is clear and the experimental results are correct. However, the phenomenon pointed out in the EV-A71 cell cultured adaptive mutants does not occur in nature host. While these mutants could be applied for vaccine development is still question.

All of the experimental results are convinced and statistical analyses are clearly shown. However, the conclusions of this manuscript may lead to a confused and it will be a fallacy for virologists

Reviewer #2: In this manuscript, Kobayashi et al. elucidated the population dynamic of EV71 during adaptation in cell culture system using NGS analysis. They found a VP1 E145G/Q substitution occurred during virus passaging in RD-A cells. This amino acid substitution is known to enhance HS binding and make virus attenuated. They also found that the population of VP1-145E virus was stable by culturing in HS deficient and hSCARB2 overexpressing (deltaEXT+hSCRB2) RD-A cells; however, VP1-145G and Q viruses gained mutations in both RD-A and deltaEXT+hSCRB2 cells. The VP1-145G/Q mutations further promoted to acquire secondary amino acid substitutions on P1 region, and these mutations are near the 5-fold axis, which is critical for virus binding to cellular receptors. VP1-145G/Q viruses had higher infectious efficiency compared with VP1-145E virus, and the infectious efficiency in deltaEXT+hSCRB2 cells was higher among VP1-145E, G and Q viruses compared to that of RD-A cells. Altogether, the authors explore a possible mechanism of EV71 adaptation and attenuation in cell culture system. The attachment receptor HS, plays an important role for selection VP1-145 mutation and other accessory mutations on EV71 virus which results in avirulent and attenuated viruses.

1.

**Part II – Major Issues: Key Experiments Required for Acceptance**

Reviewer #1: 1. Only one passage: VP1145 E to G in 2716-Ymg-03 strain; while VP1-145 E to Q in Isehara strain, what is the major “pressure” to cause this mutation?

2. What is the main reason that avirulent mutants are rapidly selected in the “RD” cultured cells? How about in the other cell line? Is there any cell tropism issue to cause this phenomena?

3. VP1-145 is a very important amino acid for virulence in mice. But, what is the role in the cultured cell?

4. VP1-145G and Q mutations also modulate infection efficiency, do these mean that this amino acid will interact with many host factors?

Reviewer #2: None

**Part III – Minor Issues: Editorial and Data Presentation Modifications**

Reviewer #1: Fig 5 is difficult to understand

Reviewer #2: 1. In line 400-403, the authors claimed the difference of infectious efficiency is due to the binding ability of VP1-145E, G, Q virus to HS attachment receptor. In author’s previous research, they demonstrated VP1-145G and Q viruses had higher HS binding ability compared with VP1-145E virus using HS agarose beads. How about the binding efficiency and entry efficiency of these viruses to RD-A and deltaEXT+hSCRB2 cells?

2. Whether VP1-145E, G, Q mutations directly affect EV71 translation/replication?

3. Instead of RD-A cells, have the authors tried other cell lines infected with VP1-145E, G, Q viruses? And what are the pattern of viral population?

4. What is the number of mouse per group for testing paralysis and mortality rate?

PLOS authors have the option to publish the peer review history of their article (what does this mean?). If published, this will include your full peer review and any attached files.

Reviewer #1: No

Reviewer #2: No

---

## [Editor Report · Decision Letter 1]

23 Feb 2020

Dear Dr. Koike,

We are pleased to inform you that your manuscript 'Heparan sulfate attachment receptor is a major selection factor for attenuated enterovirus 71 mutants during cell culture adaptation' has been provisionally accepted for publication in PLOS Pathogens.

Before your manuscript can be formally accepted you will need to complete some formatting changes, which you will receive in a follow up email. A member of our team will be in touch within two working days with a set of requests.

Best regards,

Richard J. Kuhn, PhD

Associate Editor

PLOS Pathogens

Raul Andino

Section Editor

PLOS Pathogens

Kasturi Haldar

Editor-in-Chief

PLOS Pathogens

orcid.org/0000-0001-5065-158X

Michael Malim

Editor-in-Chief

PLOS Pathogens

orcid.org/0000-0002-7699-2064

The response to the reviewers' comments and concerns were addressed in this revised manuscript. The revisions are acceptable and this manuscript represents an important contribution that is suitable for publication.
---

## [Editor Report · Acceptance letter]

11 Mar 2020

Dear Dr. Koike,

We are delighted to inform you that your manuscript, "Heparan sulfate attachment receptor is a major selection factor for attenuated enterovirus 71 mutants during cell culture adaptation," has been formally accepted for publication in PLOS Pathogens.

Best regards,

Kasturi Haldar

Editor-in-Chief

PLOS Pathogens

orcid.org/0000-0001-5065-158X

Michael Malim

Editor-in-Chief

PLOS Pathogens

orcid.org/0000-0002-7699-2064